# Preparation and Properties of Blended Composite Film Manufactured Using Walnut-Peptide–Chitosan–Sodium Alginate

**DOI:** 10.3390/foods11121758

**Published:** 2022-06-15

**Authors:** Wenqi Yan, Haochen Sun, Wenxin Liu, Hao Chen

**Affiliations:** 1Marine College, Shandong University (Weihai), No. 180 Wenhua West Road, Gao Strict, Weihai 264209, China; 201900810230@mail.sdu.edu.cn (W.Y.); 201900810229@mail.sdu.edu.cn (H.S.); liuwenxin1999@163.com (W.L.); 2The Key Laboratory of Synthetic and Biological Colloids, Ministry of Education, Jiangnan University, No. 1800 Lihu Road, Wuxi 214122, China

**Keywords:** sodium alginate, chitosan, walnut peptide, composite film, layer by layer assembly

## Abstract

In this study, layer-by-layer assembly was performed to prepare sodium alginate (SA) layer and walnut-peptide–chitosan (CS) bilayer composite films. Genipin was adopted to crosslink CS and walnut peptide. The properties of walnut peptide-CS-SA composite film were determined, and the influence of material ratio on the performance of composite film was explored. According to the results, the mechanical tensile property, oil absorption property, and water vapor barrier property of the composite film were improved with the presence of genipin. Moreover, the proportion of CS and walnut peptide had significant effects on color, transmittance, mechanical properties, barrier properties, and antioxidant properties of the composite films. Among them, the composite film containing 1% (*w*/*v*) CS, 1% (*w*/*v*) walnut peptide, and 0.01% (*w*/*v*) genipin showed the best performance, with a tensile strength of 3.65 MPa, elongation at break of 30.82%, water vapor permeability of 0.60 g·mm·m^−2^·h^−1^·kPa^−1^, oil absorption of 0.85%, and the three-phase electrochemistry of 2,2-diphenyl-1-picrylhydrazyl (DPPH) radical scavenging rate of 25.59%. Under this condition, the tensile property, barrier property, and oxidation resistance of the composite film are good, which can provide a good preservation effect for food, and has great application potential.

## 1. Introduction

Traditional plastics and their products are not easy to be degraded in nature. After entering the environmental system, they can stay in the soil and water for a long time, forming “white pollution”. Incineration may also produce toxic substances, which pose a serious threat to the ecological environment [1]. The work to solve the problem of plastic pollution mainly focuses on three aspects: first, developing substitutes for plastics and their products; second, developing degradable plastics; and third, increasing the recycling of plastics and their products [2,3,4,5,6]. However, over the last decade, there has been a growing interest in the development and use of bio-based packaging materials to replace the conventional synthetic packaging materials needed to preserve and protect foods. The development of innovative edible films has emerged as a new research area in food science [7]. Edible film is made of natural edible biomacromolecules (such as protein, lipid, sugar, etc.) as raw materials, and added with edible plasticizers, crosslinking agents, and other substances, which are edible through the interaction between molecules. The selection of edible materials with good biocompatibility, or the production of packaging films for one particular application is very important to improve the quality of fresh and processed food, and increase the shelf-life of the food and its packaging. Edible films can also be used as a good carrier for functional substances, such as antioxidants, antibacterial agents, flavors, probiotics, etc. It can not only prolong the shelf-life of food, but also improve its nutritional value, and has a good application prospect [8,9,10].

SA and CS have shown great potential for use as food coatings due to their biodegradability, biocompatibility, nontoxicity, and versatile chemical and physical properties [11]. SA is one of the most versatile biodegradable polymers, which is a linear polysaccharide extracted from brown algae. The chemical structure of alginate is composed of β-d-mannuronic acid and β-l-guluronic acid [12]. On the other hand, CS is found in the shells of crustaceans, such as shrimp and crabs. In molecular terms, CS is achieved by the deacetylation of chitin [13].

Because both CS and SA have strong hydrophilicity, the performance of single film is poor. In order to improve the single-film-forming performance of CS and SA, the electrostatic reaction between CS and SA was used to directly blend the two films. However, direct blending may result in a strong reaction between -NH^3+^ on CS and -COO^−^ on SA, resulting in a water-insoluble mixture. The viscosity of the film solution is large and uneven, resulting in poor film quality [14]. In recent years, the technology of layer-by-layer (LBL) assembly is gradually emerging. Macromolecules with opposite charges are deposited in turn, and the noncovalent interactions between polymer layers can endow bilayer films or multilayer films with better mechanical properties and barrier properties [15]. This technology can overcome the above shortcomings. Different layers in CS-SA bilayer films are not simply superimposed, but intermolecular interactions occur. Under this electrostatic interaction, molecules in the films are combined [16].

Particularly, crosslinking with genipin, which is a reagent of natural origin, improves the swelling, water resistance, and mechanical properties of the films [17]. Genipin has recently been used in biomedical applications and for controlled drug release, due to its biocompatibility and low toxicity [18]. Genipin also has excellent pharmacological value, such as protecting the liver and gallbladder, anti-inflammatory, antibacterial, anti-tumor, treating gastritis, treating diabetes, etc. [19]. Studies have shown that synthetic crosslinking reagents such as glyoxal, glutaraldehyde, and epichlorohydrin are all more or less cytotoxic, and may impair the biocompatibility of chitosan [20]. In this regard, genipin extracted from gardenia fruits is obviously advantageous. As a water-soluble bi-functional crosslinker, it reacts promptly with chitosan. The resulting crosslinked products have better biocompatibility and stability for use [20,21]. The genipin-crosslinked films’ total soluble matter, water vapor permeability, tensile strength, ultraviolet and visible light barriers, and thermal properties were improved in relation to the uncrosslinked film, whereas their moisture content and microstructure were not affected, demonstrating the effectiveness of genipin on improving the properties of gelatin-based films and the promising application of films as food packaging [22,23]. However, the crosslinking agent, genipin, is seldom used in the related research of edible film at present, and the excellent crosslinking ability and biocompatibility of genipin still need to be further explored. 

Compared with walnut protein, walnut polypeptide has better solubility, emulsifying property, oil absorption, and oxidation resistance, and is more easily absorbed and utilized by the human body; walnut polypeptide has good nutrition and health care functions, but poor solubility, and is very limited in application in the food field [24]. Research has shown that walnut peptide is rich in amino acid residues such as glutamic acid, arginine, and aspartic acid, related to its antioxidant property. In addition, walnut peptide also has a certain anti-cancer performance, and has great application potential in the fields of medicine and food, and further exploration is needed to expand the scope of application of walnut peptide [25]. In this paper, walnut peptide is added into the composite film as a functional active ingredient, which enhanced the preservative effect of the composite film and also provided a new idea for the application of walnut peptide in production. 

In order to achieve better functional properties, the application of multilayers is an available way which can take advantage of the unique properties of several materials [26,27]. In this study, CS and SA were used as the main film-forming matrix, and walnut peptide was added to enhance the antioxidant performance of the composite film. Genipin was added to chemically crosslink CS and walnut peptide, and the walnut-peptide–CS–SA composite film was prepared by a layer-by-layer assembly method. The effects of the ratio of walnut polypeptide to CS on the performances of the composite film are explored, and the performances of the composite film were measured, including thickness, mechanical tensile property, color difference value, light transmittance, water vapor transmittance, oil absorption, and antioxidant property. Further, a novel composite edible film with excellent performance and green safety is prepared.

## 2. Materials and Methods

### 2.1. Materials

SA (chemically pure, viscosity (10 g/L, 20 °C) ≥0.02 Pa·s) and CS (≥80% deacetylated, viscosity 50–800 mPa·s) were purchased from Sinopharm Chemical Reagent Co., Ltd. (Shanghai, China). Walnut peptide powder (98% purity) was obtained from Ningshan Guosheng Biotechnology Co., Ltd. (Shanxi, China). Glycerol (analytically pure) and potassium bromide (analytically pure) were from Tianjin Beichen District Fangzheng Reagent Factory (Tianjin, China), and genipin (>95% purity) was from ChangSha ZhiXin Biotechnology Co., Ltd. (Changsha, China). Acetic acid (analytically pure) and hydrochloric acid (analytically pure) were from Yantai Shuangshuang Chemical Co., Ltd. (Shandong, China). Anhydrous (analytically pure) was from Tianjin Zhiyuan chemical reagent Co., Ltd. (Tianjin, China). Soybean oil was from Yihai Jiali Cereal&Oil Food Industry Co., Ltd. (Guangdong, China). DPPH (97% purity) was from Kaima biochemical Co., Ltd. (Tianjin, China). Absolute alcohol (analytically pure) was from Tianjin Fuyu Fine Chemical Co., Ltd. (Tianjin, China).

### 2.2. Preparation of Film Solutions and Film Formation

#### 2.2.1. Sodium Alginate Monolayer Film Production

SA was weighed at 1.5 g, and then slowly added into 100 mL distilled water while stirring. The solutions were stirred continuously for 1 h using a magnetic stirrer (RH-KT/C, IKA Works Guangzhou, Guangzhou, China) in a 40 °C water bath until the powders were completely dissolved. Then, 2 mL glycerol was added, stirred evenly, and made into a 100 mL film solution. The film solution was degassed by ultrasonic cleaner (KQ-300B, Kunshan Shumei Super Sound Instrument Co., Ltd., Kunshan, China) for 10 min. Then, 15 g of the solution was poured into a plastic petri dish with a diameter of 90 mm. The solution was evenly tiled and dried in a constant-temperature air-blowing drying oven at 30 °C for 24 h to form the SA monolayer.

#### 2.2.2. Chitosan–Walnut-Peptide Covalent Complex Production

A certain amount of CS powder was weighed according to the proportion in Table 1 and dissolved in 1% (*v*/*v*, %) acetic acid solution, and magnetically stirred until completely dissolved. According to the proportion in Table 1, a certain amount of walnut peptide powder was dissolved in distilled water and magnetically stirred at 60 °C for 30 min. The walnut peptide aqueous solution was mixed with CS acetic acid solution, and 2 mL plasticizer glycerol was added.

S-CG1~S-CG5 were mixed with genipin acetic acid aqueous solution, and S-C0 was the control group without genipin. HCl (1 mol/L) was used to adjust the pH of all the six kinds of mixed solution to 3.0 to prepare the 100 mL membrane solution. The solution was degassed in an ultrasonic cleaner for 10 min, and cooled to room temperature for further use.

#### 2.2.3. Walnut-Peptide–Chitosan–Sodium Alginate Composite Edible Film Production

The preparation process is shown in Figure 1. A total of 17 g CS–walnut-peptide solution was poured onto the SA monolayer that was about to be completely dried, spread evenly, and dried in a constant-temperature blast-drying oven at 30 °C for 24 h to form a film. After drying, the film was taken down and put in a dry dish (RH60%, room temperature) containing saturated KBr solution to return moisture.

### 2.3. Fourier-Transform Infrared (FTIR) Spectroscopy

The FTIR analysis was carried out using a PIKE MIRacle Universal ATR system in a Tensor 27 FTIR-ATR spectrometer (Bruker Optik GmbH, Ettlingen, Germany) to obtain the FTIR spectra of six sample films and raw material powers (SA, CS, walnut peptide, and genipin). Each sample film was cut into a small piece directedly spread on the scanning table, and each raw material powder was mixed with KBr using a hydraulic press to form a pellet. Then, the samples were kept in the sample recording chamber under reduced vacuum. The scanning and the resolution range were 500–4000 cm^−1^ and 4 cm^−1^, respectively.

### 2.4. Mechanical Test

The composite film was cut into strips of 50 mm × 10 mm, and both ends were clamped flatly on the tensile probe of the texture analyzer (Universal TA, Shanghai Tengba Instrument Technology Co., Ltd., Shanghai, China). The initial spacing was set to be 20 mm, and the tensile speed was set to be 60 mm/min. Three parallels were made for each composite film, and the maximum tensile force and elongation length of the film at break were recorded. The results were averaged. The tensile strength (TS, MPa) and elongation at break (EB, %) were calculated by following Equations (1) and (2).
(1)$$\mathrm{TS}=\frac{F}{d\times W}$$
(2)$$\mathrm{EB}=\frac{L_{1}-L_{0}}{L_{0}}\times 100\%$$
where *F* denotes the maximum tension when the film breaks, N; *d* represents the thickness of the film, mm; *W* represents the width of the film sample, mm; *L*_0_ is the length of the stretching front film, mm; *L*_1_ is the length of the stretched film, mm.

### 2.5. Chromatic Properties

The color parameters (*L*: lightness/brightness, *a*: redness/greenness, and *b*: yellowness/blueness) of the film were measured by a colorimeter (NR110, Shenzhen 3nh Technology Co., Ltd., Shenzhen, China), and the colorimeter was used after calibration by standard white film. Taking S-CG1 sample as reference, the color difference was calculated, and the average values of three composite film samples were taken. The color difference (ΔE) was calculated by following Equation (3).
(3)$$\Delta E=\sqrt{{\left(L-L_{1}\right)}^{2}+{\left(a-a_{1}\right)}^{2}+{\left(b-b_{1}\right)}^{2}}$$
where *L*, *a*, and *b* are the color parameter values of film samples; *L*_1_, *a*_1_, and *b*_1_ are the color parameter values of the control film (S-CG1).

### 2.6. Transmittance

The light transmittance was measured by UV-visible spectrophotometer (T6, Beijing Persee General Instrument Co., Ltd., Beijing, China). The edible film was cut into 1.2 cm × 4 cm samples, which were closely attached to the inner side of the cuvette (using an empty cuvette as the control). The light transmittance of the samples was measured at the wavelength of 560 nm, and three parallel samples were measured, and the average value was taken [28]. The transmittance (T, %) was calculated by following Equation (4).
(4)$$T=0.1^{A_{560}}\times 100\%$$

### 2.7. Water Vapor Permeability (WVP)

Anhydrous CaCl_2_ (3 g) was added to a 40 mm × 25 mm weighing bottle, which was then sealed with a film, and the initial mass of the weighing bottle was recorded. The weighing bottle was then placed in a drying dish containing a saturated KBr solution at room temperature and 60% relative humidity, and the mass of the weighing bottle was measured every 24 h until the weighing bottle weight did not change, and the final mass was recorded [29]. The WVP was calculated by following Equation (5).
(5)$$\mathrm{WVP}=\frac{\Delta m\times d}{At\Delta P}$$
where WVP is the water vapor permeability of the film, g·mm·m^−2^·h^−1^·kPa^−1^; Δ*m* is the added mass of the weighing bottle, g; *A* is the surface area of the composite film, m^2^; *t* is the interval time, h; *d* is the film thickness, mm; Δ*P* is the vapor pressure difference on both sides of the composite film, kPa.

### 2.8. Oil Absorption

The film was cut into a square with a size of 25 mm × 25 mm, weighed, and laid flat on the filter paper. Then, 4 mL of soybean oil was poured onto the film and allowed to stand for 0.5 h. Oil on the surface of the dry film was absorbed by the filter paper, and the film weight was measured. The oil absorption of the film was expressed as the ratio of the increment of the film weight to the original weight of the film. The oil absorption was calculated by following Equation (6).
(6)$$\mathrm{Oil}\;\mathrm{absorption}=\frac{m_{2}-m_{1}}{m_{1}}\times 100\%$$
where *m*_1_ is the mass before oil absorption of the film, g; *m*_2_ is the mass of the film after oil absorption, g.

### 2.9. Antioxidant Activity

The film sample (0.3 g) was weighed into 30 mL of distilled water, and after 24 h, when it was dissolved completely, 30 mL of ethanol was added. The composite film solution was centrifuged at 4000 r/min for 20 min, and the supernatant was extracted as the film extract. First, 1 mL of the supernatant was mixed with 5 mL of 0.01 mmol/L DPPH ethanol solution, and the mixture was put in the dark for reaction for 30 min. The absorbance, *A_i_*, of the solution at 517 nm, and the absorbance, *A_j_*, of 1 mL film extract and 5 mL ethanol at the same time, as well as the absorbance, *A_o_*, of sample solvent (0.5 mL distilled water + 0.5 mL ethanol) were measured after standing in the dark for 30 min with 5 mL 0.01 mmol/L DPPH ethanol solution [30]. The antioxidant activity was shown as DPPH radical clearance, and it was calculated by following Equation (7).
(7)$$\mathrm{DPPH}\;\mathrm{free}\;\mathrm{radical}\;\mathrm{scavenging}\;\mathrm{rate}\%=\frac{A_{o}-A_{i}-A_{j}}{A_{o}}\times 100\%$$

### 2.10. Statistical Analysis

In the experiment, three groups of parallel samples were set for each sample, and the data were processed by Microsoft Excel 2010. The significant difference between the different samples was evaluated by Duncan analysis in SPSS 21 (*p* < 0.05). The images were drawn by Origin 2019 and KingDraw 3.0.

## 3. Results and Discussion

### 3.1. FTIR Analysis

The FTIR-ATR technique was used to evaluate the functional groups of the materials, and to detect possible changes in the proportion of raw materials. As an excellent natural biological crosslinking agent, genipin can not only improve the swelling, water resistance, and mechanical properties of the films [17], but also chemically crosslinks CS and walnut peptide and improves the density of the membrane structure [31,32]. FTIR-ATR spectroscopy was used for the analysis of intermolecular interactions between the genipin, CS, and walnut peptide, which could be reflected in infrared spectra by the location and intensity of characteristic absorption bands, as shown in Figure 2a,b.

A strong and broad adsorption band at around 3500–3000 cm^−1^ corresponded to -OH stretching, which was affected by the intermolecular or intramolecular hydrogen bonds [33]. The band corresponding to asymmetric stretching vibrations of the methylene group occurred at approximately 2950–2900 cm^−1^ [34]. In addition, the absorption peaks at the range of 1030–1010 cm^−1^ correspond to the stretching of CH_2_-OH, respectively [35].

In Figure 2a, both CS, with a degree of deacetylation of 85%, and walnut peptide displayed two strong vibrations at 1652–1648 and 1580–1550 cm^−1^. These have previously been assigned to amide I and amide II vibrations [36]. Besides, they had N-H and C-N absorption peaks at 1580 and 1324 cm^−1^ [37]. The carboxylate anion –COO^−^ of SA had two characteristic absorption peaks at 1632 and 1408 cm^−1^, and the ester bond COO of genipin also had two characteristic absorption peaks at 1661 and 1428 cm^−1^, corresponding to the stretching vibrations of C=O and C-O, respectively [38,39]. In Figure 2b, the band at 1661 cm^−1^, which represents the carbonyl group of ester in genipin, shifted to a lower frequency that appeared at 1638–1642 cm^−1^ in crosslinked films. This is due to the crosslinking reaction between genepin and CS, that is, the nucleophilic attack by the amino group of CS on the olefinic carbon atom of genipin, resulting in the opening of the dihydropyran ring and the formation of tertiary amine. The subsequent slower reaction is the formation of amide through the reaction of the amino group on chitosan with the ester group of genipin [35,40]. Walnut peptides also have many amino groups that can react with genipin. Furthermore, the increase observed in the peaks at around 1411 cm^−1^ can be assigned to absorptions from C-N stretching vibrations, respectively, more numerous after crosslinking with genipin [41]. In addition, the absorption peak at 1028 cm^−1^ could be attributed to the interaction between the -OH group of glycerol (used as a plasticizer) and genipin in all the film samples [42]. This is similar to the results of Kumar et al. [43].

### 3.2. Mechanical Properties

The composite films were successfully prepared, and the average thickness of each sample was characterized. The results showed that with the increase of walnut peptide content, the thickness of the films gradually increased from 0.1505 mm to 0.2133 mm, and the structure of composite film became loose. In order to evaluate the mechanical properties of composite films with different raw material ratios, the TS (tensile strength) and EB (elongation at break) of composite films were measured. It can be seen from Figure 3 that the TS of the composite film first increased and then decreased with the decreasing mass ratio of CS/walnut peptide in the sample added with the crosslinking agent, genipin. The changes of TS and EB were likely due to the micro network structure and intermolecular force in the film matrix formed by the crosslinking of CS, walnut peptide, and genipin [44].

TS is the maximum tensile strength that a film can sustain, thus it expresses the maximum stress developed in the film during the tensile testing [45]. The TS of the S-CG3 sample with a CS/walnut peptide mass ratio of 1:1 was the highest (3.65 MPa), which was significantly higher than that of S-C0 sample without genipin. This is because the crosslinking effect of genipin made the covalent crosslinking between CS and walnut peptide form a Schiff base [46], which made the film structure closer than CS-SA film and walnut peptide-SA film, and it could better resist the tearing of external forces. However, when the walnut peptide ratio was too large, the strong interaction between CS and SA was replaced by the weak interaction between peptides, and the tensile properties of the bilayer film were also affected. Moreover, too high amounts of walnut peptides possibly caused inhomogeneity of the polymer matrices, which subsequently reduced the mechanical properties [47].

EB is a measure of the film’s capacity for stretching, and also refers to the maximum change in length of the test specimen before breaking [48]. Elongation of the films depended on several factors, including homogeneity of the matrices and plasticization, which increased the flexibility and deformability of the polymers [49]. The elongation at break of the composite film first increased and then decreased from S-CG1 to S-CG4. The binding sites of CS and walnut peptides were limited, so when the proportion of walnut peptides gradually increased, the binding between the two tended to be saturated [50]. The crosslinking agent, genipin, made the film dense, which made the film not easy to be stretched, so the elongation at break of the S-CG3 sample was smaller than that of the S-C0 sample without crosslinking. When the upper film did not contain CS, the polypeptide structure of walnut peptide was loose, which made the film sticky and flexible, and caused maximum elongation at break. 

### 3.3. Chromatic Properties

The measurement results of the chromaticity values, *L*, *a*, and *b*, of each composite film sample and the color difference, ΔE, of S-CG1 to S-CG5 samples are shown in Figure 4. The addition of plant-derived materials commonly gives color to the composite films, thereby affecting the light transmission [51]. Genipin can react with amino acid or proteins to form dark blue pigments [52]. The color of the composite film without adding genipin was pale yellow, and that of the composite film with adding genipin was blue-green, which was due to the gardenia blue pigment generated by the reaction of genipin with the amino groups of CS and walnut peptide [46]. 

The reaction mechanism is shown in Figure 5. The more adequate their action, the more pigment generated, and the deeper the blue color. The results showed that with the increase of the proportion of walnut peptide, the *L*, *a*, and *b* values of the composite film fluctuated, and the ΔE values of S-CG2-S-CG5 samples were higher, indicating that, compared with the S-CG1 sample without walnut peptide, it had a significant color difference. In summary, the reaction mode and reaction adequacy between walnut peptide and genipin or CS greatly affected the color of the composite film.

### 3.4. Transmittance Properties

The transmittance of composite film can reflect the compatibility of film components. If the transmittance of composite film is very low, the material compatibility is poor. As shown in Figure 6, the transmittance of the S-C0 sample without genipin crosslinking was the highest, which was 85.06%. Pigments were generated in samples S-CG1~S-CG5 due to the crosslinking reaction of genipin, resulting in a decrease in the transmittance of the film, which fluctuated between 31.86% and 42.38%. This was due to the light reflection and scattering at the interface of the continuous phase and the dispersed phase when CS and walnut peptides were combined, resulting in changes in the transmittance. Moreover, increased filler contents possibly filled up the void volume between polymer molecules, and hence, blocked the transmission path of the light [53]. The mixed systems of different layers of the composite film have heterogeneity, which is also the reason for the decrease of transmittance.

### 3.5. Water Vapor Transmission Rate

Water vapor permeability is an important index to measure the barrier performance of film. Low water vapor permeability is beneficial to prevent water exchange between food and environment in wet environments. The measurement results of water vapor permeability of each composite film sample are shown in Figure 7. Among them, the water vapor permeability of S-CG1 and S-CG3 samples was the lowest, which was 0.60 g·mm·m^−2^·h^−1^·kPa^−1^, indicating that the CS-SA bilayer structure and CS- walnut peptide crosslinking system could play a role in blocking water. S-CG1, S-CG2, and S-CG3 showed low WVP; it is possible that at lower amounts of walnut peptides, the film matrices were denser, which limited the diffusion of water vapor, giving lower WVP [54]. Walnut peptides have strong hygroscopicity. When the proportion of walnut peptide increased, the water vapor permeability increased; it is probable that the walnut peptide and CS could not be fully crosslinked, or high amounts of walnut peptides caused non-homogeneous structures and disrupted networks [55].

### 3.6. Oil Absorbency Capacity

It can be seen from Figure 8 that the oil absorption of the composite film first decreased and then increased with the increase of the walnut peptide addition ratio. After the addition of genipin, the oil absorption of the film decreased, and the oil absorption of the S-CG3 sample was the weakest, which was 0.85%. This is because the CS and walnut peptide crosslinked to form a dense network structure, blocking the oil into the film, which, to a certain extent, weakened the film oil absorption. However, walnut peptide containing a high content of hydrophobic amino residues has high oil absorption, so when the proportion of walnut peptide increased to a certain range, the oil absorption of the film increased [56]. In addition, with the increase of the walnut-peptide–chitosan ratio, the non-homogeneous structures and void formation were formed, which also promoted the improvement of oil absorption.

### 3.7. Antioxidant Activity

The oxidation resistance of the composite film is an important standard to measure the effect of packaging preservation. Previous studies have shown that walnut polypeptides are macromolecular substances synthesized by amino acids such as glutamic acid, arginine, and aspartic acid. The molecular structure of walnut polypeptides contains plenty of amino groups, which have DPPH free radical scavenging activity and ACE inhibitory activity [57,58]. Besides, CS has active hydroxyl groups, indicating stronger DPPH free radical scavenging ability [59,60,61]. In addition, the layer of SA, with certain antioxidant activity, can also improve the scavenging ability of composite films [62]. 

In the experiment, walnut peptide was added to endow the composite film with oxidation resistance, and the DPPH radical scavenging ability of the composite film samples was measured to evaluate the oxidation resistance of the composite film. As can be seen in Figure 9, with the increase of the walnut peptide addition ratio, the DPPH radical scavenging rate of the film increased gradually, and the highest scavenging rate of the S-CG5 sample was 30.14%. However, the S-CG1 sample without walnut peptide also showed a certain antioxidant capacity, because CS also had a certain antioxidant capacity, and its antioxidant capacity was related to the degree of deacetylation [63]. In summary, walnut peptides could improve the oxidation resistance of composite films.

## 4. Conclusions

Layer-by-layer assembly was performed to prepare SA layer and walnut-peptide–CS layer bilayer composite films. Genipin crosslinked CS and walnut peptide to form a network structure that can improve the mechanical strength of the composite film. In addition, the tensile property, barrier property, and oxidation resistance of the composite film are good. The appropriate ratio of CS and walnut peptide can create the crosslinking reaction more fully and improve the tensile strength and elongation at break of the film. The pigments generated by the crosslinking reaction of genipin could affect the color and optical properties of the film. The crosslinking of CS and walnut peptide by genipin makes the film dense, and oil and water cannot easily penetrate the composite film, which is conducive to the preservation of food. However, when the walnut peptide ratio increases, its moisture absorption and oil absorption will affect the barrier performance of the film. The addition ratio of walnut peptides was positively correlated with the DPPH free radical scavenging rate of the composite film. The walnut peptides endowed the composite film with antioxidant capacity, so that the composite film had better preservation function. Comprehensively, the composite film with a CS/walnut peptide ratio of 1:1 and genipin addition had the best performance. The tensile strength was 3.65 MPa, the elongation at break was 30.82%, the water vapor permeability was 0.60 g·mm·m^−2^·h^−1^·kPa^−1^, the oil absorption was 0.85%, and the DPPH radical scavenging rate was 25.59%. The composite film had good mechanical properties and barrier properties, which could provide good preservation effects.

## Figures and Tables

**Figure 1 foods-11-01758-f001:**
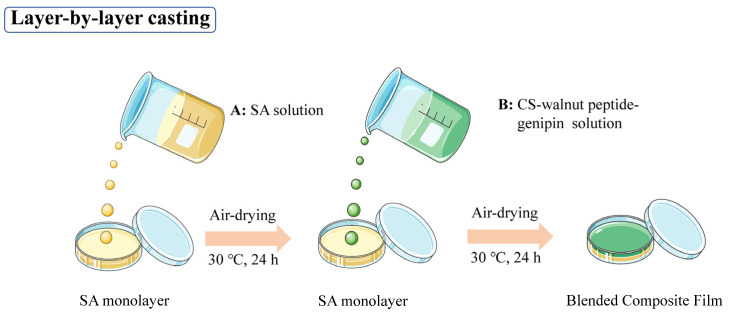
The preparation process of films.

**Figure 2 foods-11-01758-f002:**
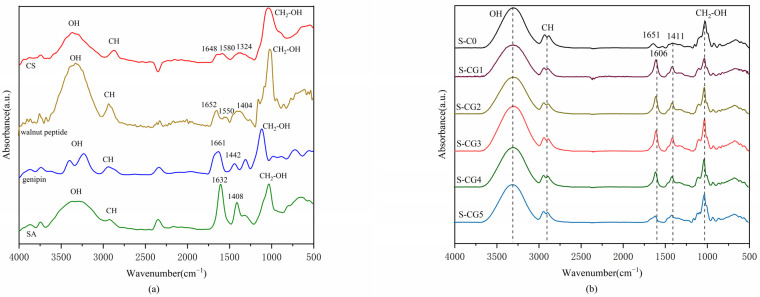
FTIR spectra of raw material (**a**) powders, including CS, walnut peptide, genipin, and SA, and six sample films described in Table 1 (**b**).

**Figure 3 foods-11-01758-f003:**
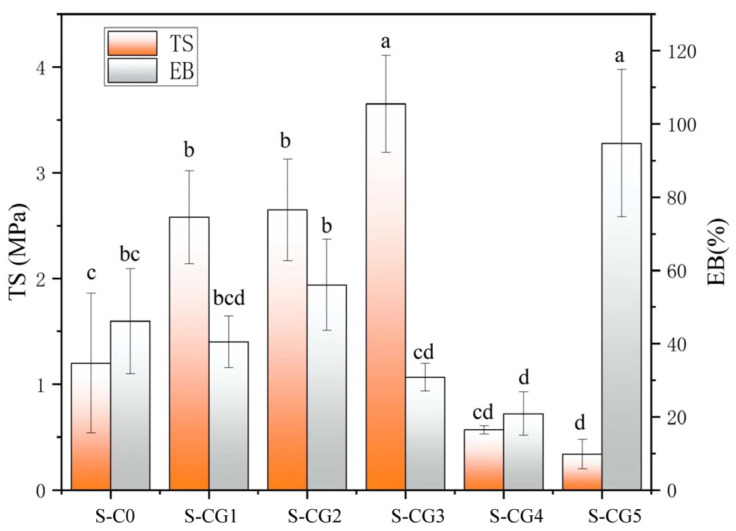
The tensile strength and elongation at break of composite films. Results are expressed as average, and different letters (a–d) indicate significant differences (*p* < 0.05).

**Figure 4 foods-11-01758-f004:**
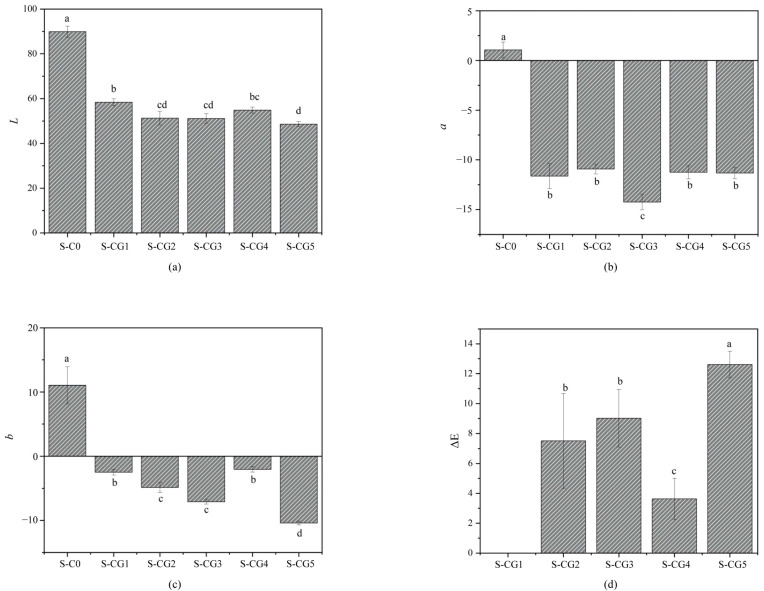
The chroma values of composite films. (**a**) *L*: lightness/brightness, (**b**) *a*: redness/greenness, (**c**) *b*: yellowness/blueness, (**d**) ΔE: the color difference. Results are expressed as average, and different letters (a–d) indicate significant differences (*p* < 0.05).

**Figure 5 foods-11-01758-f005:**
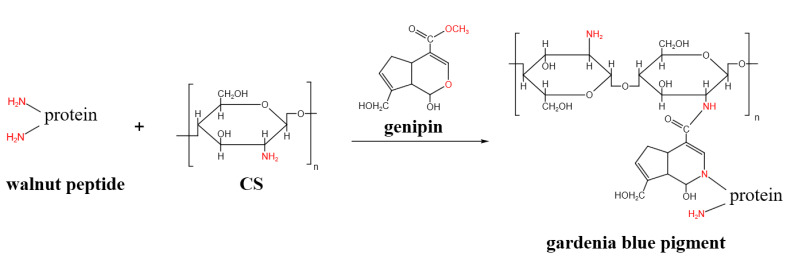
Proposed mechanism of genipin reaction with CS and walnut peptide to produce gardenia blue pigment.

**Figure 6 foods-11-01758-f006:**
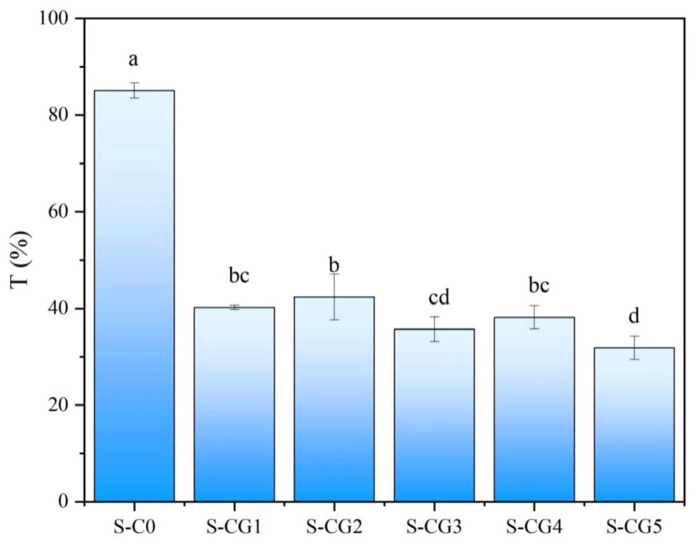
The transmittance of composite films. Results are expressed as average, and different letters (a–d) indicate significant differences (*p* < 0.05).

**Figure 7 foods-11-01758-f007:**
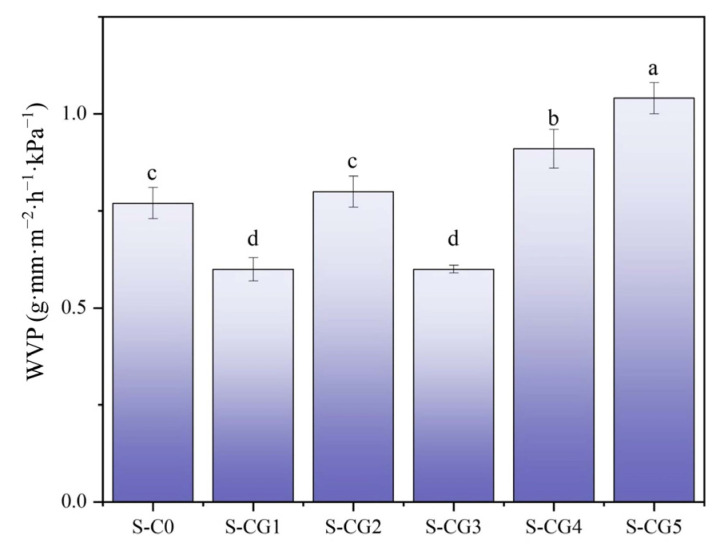
The water vapor permeability of composite films. Results are expressed as average, and different letters (a–d) indicate significant differences (*p* < 0.05).

**Figure 8 foods-11-01758-f008:**
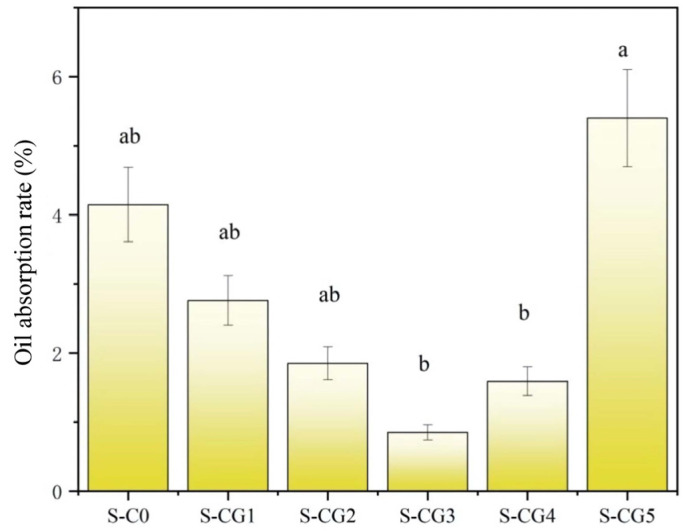
The oil absorption of composite films. Results are expressed as average, and different letters (a, b) indicate significant differences (*p* < 0.05).

**Figure 9 foods-11-01758-f009:**
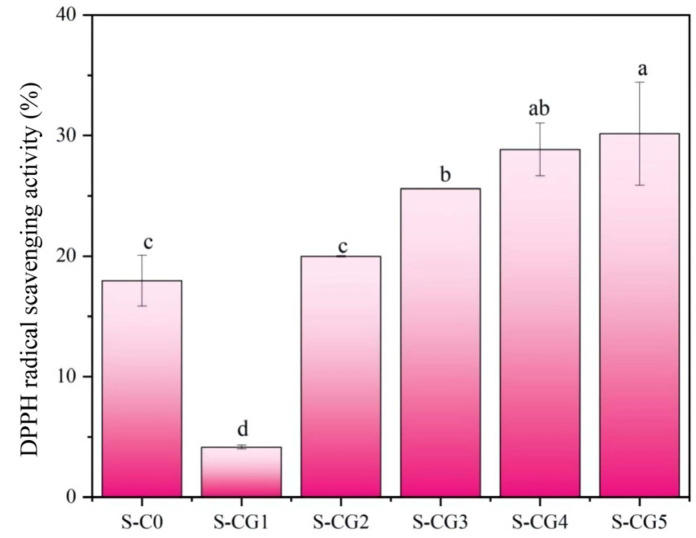
The DPPH free radical scavenging activities of composite films. Results are expressed as average, and different letters (a–d) indicate significant differences (*p* < 0.05).

**Table 1 foods-11-01758-t001:** Composition of different composite films.

Sample Number	SA (*w*/*v*, %)	Glycerol (Lower Layer) (*v*/*v*, %)	CS (*w*/*v*, %)	Walnut Peptide (*w*/*v*, %)	Glycerol (Upper Layer) (*v*/*v*, %)	Genipin (*w*/*v*, %)
S-C0	1.5	2.0	1.0	1.0	2.0	0
S-CG1	1.5	2.0	2.0	0	2.0	0.01
S-CG2	1.5	2.0	1.5	0.5	2.0	0.01
S-CG3	1.5	2.0	1.0	1.0	2.0	0.01
S-CG4	1.5	2.0	0.5	1.5	2.0	0.01
S-CG5	1.5	2.0	0	2.0	2.0	0.01

## Data Availability

The data are available from the corresponding author.

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
