# Peer review of "Preparation and Properties of Blended Composite Film Manufactured Using Walnut-Peptide–Chitosan–Sodium Alginate"

_foods, 2022, doi:10.3390/foods11121758_

Round 1
Reviewer 1 Report
Yan et al. fabricated the blended composite film using walnut peptide/chitosan/sodium alginate.
Authors claimed that the prepared composite films developed through by layer by layer process.
However, in the experimental section it is not clear please clarify it the sample preparation method.
The present manuscript has not details in methods part.
The prepared films have not detected by FTIR and NMR. These data must include in the manuscript. At least FTIR data is necessary.
Line 18, please write superscript clearly. (g·mm·m-2·h-1·kPa-11)
Additional experiment must be need XRD and SEM of cross section. SEM also need after mechanical properties testing.
Why Genipin is selected in this study as an crosslinkers only to increase antioxidant activity. There are other cross linker available which can increase the antioxidant activity. Make the justification
and compare other crosslinkers effect with this study.
Authors need to provide a reaction mechanism scheme among the raw materials.
Ref [1] is cited after full stop. Please check it same errors throughout the manuscript.
Line 44. Must have space in between reference and word.
Define Chitosan as short name and used it throughout the manuscript. SA some time authors using short name and full name. Once it is defined in short name after that should be short name
throughout the manuscript.
Line 117, A certain amount of chitosan powder was weighed according to the proportion in 117 Table 1 and dissolved in 1% (v/v, %) acetic acid solution, and magnetically stirred until completely
dissolved” what is conc. of chitosan. Not define in the table.
In layer by layer method how many layer authors prepared. What is the thickness of film? All film thickness is same? Did authors measure its density. All the basic information’s should be included
in the manuscript. What is crosslinking density of all prepared samples?
Check table 1, seems not right informations.
Section 2.2 is not written well. Please rewrite clearly how authors prepare solution and then describe film preparation details clearly.
Line 116, space should be between number and ℃ “oven at 30℃ for 24 h to form sodium alginate monolayer”. Line 108 also 1.5g. Check these types of errors throughout the manuscript.
What is a, b, c and L1, a1 and b1 please define it clearly in the section 3.
Line 145 and 146 clearly write subscript in L0 and L1. For oil absorbency contact angle data is necessary in order to prove the hydrophobic material.
Very less references are cited for this manuscript. Here few references are suggested and indicated in the text accordingly. Food Bioscience, 2022, 46, 101570, Food Chemistry, 2019, 283, pp. 397–403
In all figure what is a b c d. indicate which one is significant different to which letters.
Conclusion section need to be shorten. Please remove bullet point. Please make it precisely.
All authors are belonging to one affiliation; there is no meaning indicate 1. Check it in affiliations sections.
Section 2.1, all sentences are not complete. It is not correct way to provide information. Please re-construct new sentences.
Line 214- 221 need to rewrite in better way.
Full scan UV spectra need to provide in the manuscript.
In antioxidant activity, all film is soluble in the solvent. What the mechanism of the antioxidant film for this study. Need to present it. What is the antioxidant activity of pure chitosan and alginate film?
DPPH full name is missing in the manuscript.
Line 111, 2ml, should be written as 2 mL. Check and correct throughout the manuscript.
Figure 2, subfigure label a, b, c and d need to write inside the figures. and need to mention what is a b c and d in the fig.2 caption.
There are no references in the result and discussion. All section need references to support conclusions.
Reviewer 2 Report
The manuscript provides interesting results on the biopolymer packaging which is the research interest by global researchers. Here are the comments that need to be reconsidered.
Recheck and add the space between number and abbreviation “g”, “ml”, “h” and etc.
Table 1 What are the “Glycerol (lower layer)”? Why were there two with different units?
Perhaps, “membrane” should be reconsidered as “film” for the whole manuscript to be consistent and comply with the title.
L178 should be “where…”
L202 Revise the English writing
L207 Any confirmation/citation about the cross-linking?
L213 Add more discussion and citation e.g. Moreover, too high amounts of walnut peptides possibly caused inhomogeneity of the polymer matrices which subsequently reduced mechanical properties (https://doi.org/10.1002/app.45533).
L221 Does this mean flexible and fracture at the same time? Is it possible?
Add more discussion and citation e.g. Elongation of the films depended on several factors including homogeneity of the matrices and plasticization which increased flexibility and deformability of the polymers (https://doi.org/10.1016/j.foodchem.2021.130956).
L226 Add more discussion and citation e.g. Addition of plant derived materials commonly gave color to the composite films which further affected light transmission (https://doi.org/10.1016/j.lwt.2021.112356).
L228 Did the color attribute to the color of filler itself or the reaction? Any proof of the reaction?
L247 Add more discussion and citation e.g. Moreover, increased filler contents possibly filled up the void volume between polymer molecules and hence blocking the transmission path of the light (https://doi.org/10.1016/j.fpsl.2020.100521).
WVP Add more discussion why the results are fluctuated e.g., It is possible that at lower amounts of walnut peptides, the films matrices were denser which limited diffusion of water vapor, giving lower WVP (https://doi.org/10.1016/j.foodchem.2021.131709).
L258-261 Recheck English
L261 High amounts of walnut peptides caused non-homogeneous structures and disrupted networks which possibly facilitated water vapor diffusion and increased WVP value (https://doi.org/10.1016/j.foodcont.2021.108541).
Fig.5 Recheck the stat labels.
L270-271 Was it hydrophobic? Is it possible that the non-homogeneous structures and void formation allowed for higher oil absorption (e.g. increased surface area and volume)?
Conclusion is too long. Avoid repeating the results.
Round 2
Reviewer 1 Report
The authors revised the manuscript well. However, in Fig. 1, Y-axis is not Transmittance (%), it is absorbance (a.u.). Please double-check and correct it. FTIR (Fig. 2) has only two spectra, but the authors prepared 5 samples as per Table 1, so please present 5 samples of spectra with raw materials (SA, CS, and walnut) and compare each other and analyzed them. It is the main part of the sample preparation confirmation. Without this, it's impossible to know interaction. CS (Molecular weight and DD) and SA (Molecular weight) materials information is missing. Please mention this information in section 2.1. Section 3.9, Line 604-606, "However, the S-CG1 sample without walnut peptide also showed certain antioxidant capacity, because CS also had certain antioxidant capacity, and its antioxidant capacity was related to the degree of deacetylation", This line needs reference International journal of Biological macromolecules 136 (2019) 661-667. References styles are wrong. Reference 1-15 missing journal name. The authors mention some references have journal full names and some abbreviations. please follow the journal guidelines and put the journal name accordingly. Forex. ref [16] food Biosci. (here author abbreviation used) and Ref [20,21], (here authors used the full name of the journal). Journal abbreviation needed by "Polymers" journal. Check all references. Check all references accordingly. All Figure's clarity is so low, not visible clearly. Please improve it. Write one abbreviation of in manuscript. For ex: once you define FTIR in manuscript, please do not use full name again in manuscript. Just use FTIR in whole manuscript. Check other abbreviation in whole manuscript.
Reviewer 2 Report
The manuscript has been revised as recommend.
Author Response
Thank you for your kind suggestion and approval.